# Comparison of Intravenous Microdialysis and Standard Plasma Sampling for Monitoring of Vancomycin and Meropenem Plasma Concentrations—An Experimental Porcine Study

**DOI:** 10.3390/antibiotics12040791

**Published:** 2023-04-21

**Authors:** Johanne Gade Lilleøre, Sofus Vittrup, Sara Kousgaard Tøstesen, Pelle Hanberg, Maiken Stilling, Mats Bue

**Affiliations:** 1Department of Clinical Medicine, Aarhus University, 8200 Aarhus, Denmark; 2Aarhus Denmark Microdialysis Research Group (ADMIRE), Aarhus University Hospital, 8200 Aarhus, Denmark; 3Department of Orthopedic Surgery, Aarhus University Hospital, 8200 Aarhus, Denmark

**Keywords:** microdialysis, vancomycin, meropenem, pharmacokinetics, plasma sampling

## Abstract

Microdialysis is a catheter-based method suitable for dynamic sampling of unbound antibiotic concentrations. Intravenous antibiotic concentration sampling by microdialysis has several advantages and may be a superior alternative to standard plasma sampling. We aimed to compare concentrations obtained by continuous intravenous microdialysis sampling and by standard plasma sampling of both vancomycin and meropenem in a porcine model. Eight female pigs received 1 g of both vancomycin and meropenem, simultaneously over 100 and 10 min, respectively. Prior to drug infusion, an intravenous microdialysis catheter was placed in the subclavian vein. Microdialysates were collected for 8 h. From a central venous catheter, plasma samples were collected in the middle of every dialysate sampling interval. A higher area under the concentration/time curve and peak drug concentration were found in standard plasma samples compared to intravenous microdialysis samples, for both vancomycin and meropenem. Both vancomycin and meropenem concentrations obtained with intravenous microdialysis were generally lower than from standard plasma sampling. The differences in key pharmacokinetic parameters between the two sampling techniques underline the importance of further investigations to find the most suitable and reliable method for continuous intravenous antibiotic concentration sampling.

## 1. Introduction

Microdialysis is the current preferred method for continuous monitoring of antibiotic concentrations in various tissues [1,2,3,4]. Lately, microdialysis has been developed for intravenous use allowing for dynamic sampling of unbound antibiotic concentrations in plasma as an alternative to standard plasma sampling [5,6].

Continuous intravenous antibiotic concentration sampling by microdialysis has numerous advantages. First, the intravenous microdialysis catheter can stay in the peripheral venous system for up to three days [7]. This minimizes pain and discomfort compared with repeated plasma sampling and removes the risk of sampling anemia in patients (e.g., children) or small research animals (e.g., rats and mice) [5,6,8,9]. Second, intravenous microdialysis performs a selective membrane-specific exclusion of high-molecular substances (e.g., proteins and microorganisms) providing immediate availability of the unbound pharmacologically active fraction of a drug and reduces the risk of post-sampling degradation and transmission of blood-borne diseases [8,10,11]. Third, intravenous microdialysis allows for simultaneous sampling from several investigation sites, which makes comparisons between tissue compartments easy and more valid.

Application of personalized medicine through therapeutic drug monitoring of antibiotic concentrations is associated with improved clinical outcomes, especially for treatment of complex bacterial infections [12,13,14]. Therapeutic drug monitoring is not relevant in all clinical situations; nonetheless, it may be crucial for certain patient groups and drug types. All types of antibiotics have a defined pharmacokinetic/pharmacodynamic plasma treatment target, which has been evaluated through continuous assessment of plasma concentrations, conventionally based on standard plasma samples [5,15]. However, given its advantages, intravenous microdialysis may be suitable as a preferable sampling method for therapeutic drug monitoring [3]. Until now, very few studies have applied intravenous microdialysis for assessment of vancomycin and meropenem plasma concentrations.

In this porcine study, we aimed to compare the free, unbound concentrations of both vancomycin and meropenem obtained by continuous intravenous microdialysis sampling and by standard plasma sampling.

## 2. Results

All eight pigs completed the study, and data were obtained from all intravenous microdialysis catheters. Relative recovery ranged from 41 to 64% for vancomycin and from 55 to 77% for meropenem.

### 2.1. Vancomycin

A higher area under the concentration/time curve (AUC_0–8h_), peak drug concentration (C_max_), and a faster time to reach C_max_ (T_max_) were found in the standard plasma samples compared to intravenous microdialysis samples. No difference was observed for half-life (T_1/2_) between the two sampling methods (Table 1). Mean concentration–time profiles and individual concentration–time profiles for vancomycin are presented in Figure 1a,b.

### 2.2. Meropenem

Higher AUC_0–8h_ and C_max_ and a longer T_1/2_ were found in standard plasma samples compared with intravenous microdialysis samples. No difference was found for T_max_ between the two sampling methods (Table 2). Mean concentration–time profiles and individual concentration–time profiles for meropenem are presented in Figure 2a,b.

## 3. Discussion

This study investigated and compared antibiotic concentrations obtained by continuous intravenous microdialysis sampling and standard plasma sampling, for co-administered vancomycin and meropenem during an 8 h interval. The main finding was differences in key pharmacokinetic parameters between the two sampling techniques. Both vancomycin and meropenem concentrations obtained with intravenous microdialysis were generally lower than from standard plasma sampling, particularly in the distribution phase, illustrated by higher plasma C_max_.

Intravenous microdialysis has previously been used to evaluate metabolites (e.g., lactate, glucose, pyruvate, and creatinine) [8,16,17,18,19,20], antibiotics (e.g., glycopeptides, aminoglycosides, and cephalosporins) [5,6,21], and antifungals (e.g., fluconazole) [22]. Studies investigating metabolites have primarily reported lower metabolite concentrations obtained by intravenous microdialysis in comparison to standard plasma sampling [16,17,18,19,20], although one study found a good correlation [21]. For flomoxef and fluconazole, comparable concentrations were found with the two sampling methods [21,22]. For ceftriaxone, a clinical study on healthy volunteers investigated protein binding between intravenous microdialysis and plasma sampling and reported differences; it introduced a discussion of which method is most suited to capture dynamic protein binding of highly bound proteins [23].

For vancomycin, two studies, one in vitro and one in vivo, have evaluated the feasibility of intravenous microdialysis [5,6]. The in vitro study used human plasma mixed with fixed vancomycin concentrations to evaluate the feasibility of intravenous microdialysis sampling [5]. Compared to our study, they found a higher relative recovery (91% ± 3.7%), and concluded good feasibility, as a steady relative recovery was found. The apparent differences between the in vitro study design and the present porcine in vivo design may readily explain the different findings. The in vivo study conducted a clinical comparison of intravenous microdialysis and standard plasma sampling and found a good alignment between the two sampling methods [6]. In contrast to our study, vancomycin was administered as monotherapy, and a human albumin 1% solution was used as the microdialysis perfusate [6]. Whether these differences can explain the different results remains to be investigated. Interestingly, a great inter-patient variability in microdialysates compared to standard plasma samples was found. The authors conclude that intravenous microdialysis sampling is inadequate for replacement of standard plasma sampling for vancomycin.

For meropenem, two in vivo studies have evaluated the pharmacokinetics by intravenous microdialysis in a rat and rabbit model, respectively [24,25]. Both experimental models are restricted by limited plasma volume, and thus no comparable standard plasma sampling was performed. Moreover, a different infusion time and dosage of meropenem were applied, as well as different microdialysis membranes, making it difficult to compare our findings to these studies. No studies comparing intravenous microdialysis and standard plasma sampling for meropenem have been performed until now.

Despite a similarity in the course of the concentration–time profiles for vancomycin and meropenem in the present study, generally lower concentrations were found for intravenous microdialysis sampling compared to standard plasma sampling. This may be explained by different factors. (1) Microdialysis settings: when using microdialysis, different methodological equipment settings must be considered, such as perfusion fluid, flow rate, and membrane length (surface area). All these factors may affect the precision of the obtained concentrations. For intravenous microdialysis sampling, it has been recommended to apply a flow rate of 0.3 μL/min and a membrane length of 30 mm to increase relative recovery [6,7,19,26]. In microdialysis studies, it is generally recognized that an acceptable trade-off between the ideal setup and the experimental requirements is unavoidable. We applied the longest membrane length (30 mm) possible, but due to the half-life of the investigated drugs, a flow rate of 1 μL/min was chosen to ensure sufficient temporal resolution. (2) Physiological and anatomical factors: the flow rate in the blood circulation (outside the membrane) is much higher than the flow rate inside the microdialysis membrane, which may comprise the quality of the diffusion across the semipermeable membrane. It has been recommended to implant the microdialysis catheter into a blood vessel with a diameter of >3 mm to prevent blocking of the microdialysis membrane and make recovery less sensitive to alterations in blood flow [19,27]. However, this may be practically challenging in a clinical setting. Moreover, the accuracy of measured concentrations for antibiotics with a short half-life, such as meropenem, may be more exposed and affected when evaluated with intravenous microdialysis in a high-perfused organ (with altering blood pressure), as diffusion across the microdialysis membrane occurs over a certain sampling interval and the obtained concentrations are attributed to the midpoint of the sampling interval. This may particularly affect the initial peak concentrations during the distribution phase due to fast drug elimination from the blood and to some extent explains our C_max_ results for meropenem (Table 2). On the other hand, standard plasma sampling and the following ultrafiltration may be too static and not encompassing all the dynamic changes during the drug distribution phase. Furthermore, we did not quantify the extent of plasma protein binding of the investigated drugs. As this fraction may differ between pigs and humans, this can affect the translational potential of our findings and should be included in future studies. In addition, we applied different analytical methods between the two drugs for the standard plasma sampling (uniform methods may be considered in future studies) and co-administered vancomycin and meropenem and thus not as monotherapy. Theoretically, co-administration could influence protein binding, the diffusion across the microdialysis membrane, and sampling of the two molecules due to potential interactions. Nonetheless, the vancomycin and meropenem setup simulates relevant treatment regimens of various infectious disease, e.g., osteomyelitis and central nervous system infections [28,29]. Finally, the applied descriptive statistics is restricted to the actual data. A future model-based approach may be employed to accurately encompass the variations and predict the effect of, e.g., different dosing regimens.

To improve clinical outcomes through augmented therapeutic drug monitoring, it is important to find the most suitable and reliable method for continuous intravenous antibiotic concentration sampling, including locating the most appropriate analytical method for quantification of unbound antibiotic concentrations. Future studies should therefore focus on investigating and optimizing every specific methodological step to increase its validity.

## 4. Materials and Methods

This study was conducted at the Institute of Clinical Medicine, Aarhus University Hospital, Denmark, conducted in accordance with ARRIVE guidelines and approved by the Danish Animal Experiments Inspectorate. It was carried out in agreement with existing laws (license no. 2017/15-0201-01184). To meet the three Rs by reducing the number of animals used, the same pigs have provided data to other studies with different purposes [29,30]. All chemical analyses were performed at the Department of Forensic Medicine, Aarhus University Hospital and at the Department Clinical Biochemistry, Aarhus University Hospital.

### 4.1. Microdialysis

Microdialysis is a catheter-based technique, consisting of a precision pump, a catheter with a semipermeable membrane at the tip, and a collecting vial. Through the semipermeable membrane, continuous diffusion occurs according to the concentration gradients allowing for dynamic sampling of unbound water-soluble antibiotics from the extracellular fluid [4,31].

In the microdialysis system, complete equilibrium will not occur across the semipermeable membrane, due to the constant flow. As such, the obtained concentration in the dialysates represents a fraction of the actual concentration. This fraction is referred to as the relative recovery, and it is essential to calculate to determine the absolute concentration of an analyte. In this study, calibration using retrodialysis by drug was used. Relative recovery was calculated by the following equation:(1)$$Relative\;recovery\;\left(\%\right)=100\;\cdot \left(1-\frac{C_{dialysate}}{C_{perfusate}}\right)$$

*C_dialysate_* and *C_perfusate_* represent the concentration of vancomycin or meropenem in the dialysate and perfusate, respectively.
(2)$$C_{plasma}=100\;\cdot \left(\frac{C_{dialysate}}{Relative\;recovery\;\left(\%\right)}\right)$$

*C_plasma_* is the absolute plasma concentration of vancomycin or meropenem, respectively.

All the microdialysis equipment was acquired from M Dialysis AB (Stockholm, Sweden), and consisted of 67 Intravenous Microdialysis Catheters with a membrane length of 30 mm, a 20 kD cut-off, and 107 Microdialysis pumps set at a flow rate of 1 μL/min.

### 4.2. Study Design, Anesthetic, and Intra-Vascular Procedure

Eight female pigs, Danish Landrace Breed, weight 78–82 kg, were included. The pigs were anesthetized by a combination of fentanyl (0.6–0.7 mg/h) and propofol (550–600 mg/h) throughout the study period. pH (range 7.40–7.55) was monitored and regulated through ventilation. Body temperature (range 36.5–39 °C) was monitored and regulated with blankets or icepacks. pH and body temperature were within a normal porcine homeostatic range. Intravenous microdialysis catheters were inserted into the left subclavian vein, guided by ultrasound and using a peripheral venous introducer. On the opposite side of the throat, a central venous catheter was placed for standard plasma sampling (Figure 3). At the end of the sampling period, all pigs were euthanized by intravenous injection of pentobarbital.

### 4.3. Vancomycin and Meropenem Administration and Sampling Procedures

After placement, the intravenous microdialysis catheters were perfused with 0.9% NaCl followed by a 20 min tissue equilibrium period. Following this, vancomycin (1 g) and meropenem (1 g) were administered simultaneously, through separate peripheral venous catheters placed in both ears of the pig, defining time 0. Vancomycin was infused over 100 min and meropenem over 10 min. Dialysates were collected during an 8 h sampling period every 30 min from time 0 to 240 min and every 60 min from time 240 to 480 min, resulting in 12 dialysates per pig. Plasma samples, each containing approximately 4 mL, were collected from the central venous catheter in the middle of every dialysate sampling interval to EDTA tubes. After the sampling period, the perfusates were changed to 0.9% NaCl fluid containing 100 μg/mL meropenem and 300 μg/mL vancomycin, allowing for calibration according to retrodialysis by drug method [31]. After a 30 min equilibrium period, one calibration sample was collected over a 40 min interval. Collected plasma samples were stored at 5 °C for a maximum of 6 h before being centrifuged at 3000× *g* for 10 min. Dialysates and plasma samples were frozen at −80 °C until analysis [30].

### 4.4. Quantification of Meropenem and Vancomycin Concentrations

The free, unbound concentrations in the plasma samples: vancomycin was quantified using a clinical standard homogeneous enzyme immunoassay method (Chemistry XPT, Advia Chemistry, Erlangen, Germany). The principle of this analysis is that (in competition) the free drug competes with a drug–enzyme conjugate for the antibody (Appendix A). For this assay, intra-run (total) imprecisions were ±1.2 μg/mL (2 SD) at 6.6 μg/mL and ±3.7 μg/mL (2 SD) at 29.1 μg/mL [33]. Meropenem was quantified using ultra-high-performance liquid chromatography (UHPLC). Quantification was performed after ultrafiltration of the plasma samples using a centrifree ultrafiltration device with a nominal molecular weight limit of 30 kDa (Millipore). Some 5 µL of the filtrate was prepared for analysis. The inter-run imprecision (percentage coefficients of variation (%CV)) was 3.0% at 2.0 µg/mL. The quantificational accuracy of meropenem was within −4.3% and 4.8%, and displayed a linearity range of 0.5 µg/mL to 105 µg/mL [34].

The free, unbound concentrations in the microdialysates: quantification of vancomycin and meropenem concentrations was performed using UHPLC and tandem mass spectrometry. Before analysis, microdialysates were prepared by a mixture of 5 µL microdialysate sample and a 300 µL internal standard solution in a 96-well microplate. As internal standard, 0.1 µg/mL norvancomycin was used for vancomycin and 0.1 µg/mL meropenem-D6 for meropenem.

For calibration, separate samples using reference compounds were prepared (Vancomycin Hydrochloride EDQM Reference Standard CRS batch 3 and Meropenem Trihydrate Ventranal analytical standard). A 3 µL sample volume was introduced into the UHPLC system with a C18 column and further analyzed with mass spectrometry. With positive electrospray ionization, the compounds were detected, using the following m/z transitions: vancomycin 725.2 → 144.1 and norvancomycin 718.5 → 144.1, meropenem 384.1 → 68 and meropenem-D6 390.2 → 147.1. Construction of calibration curves was carried out by linear regression of the peak area ratio (analyte/internal standard) versus the nominal analyte concentrations and further based on seven points (including the blank).

Acceptable levels of precision (CV < 15%), in the quantification ranges of 0.1 to 20 µg/mL, were shown for both drugs [35,36].

The lower limits of quantification for the applied analytical methods are individually depicted in Figure 1a and Figure 2a.

### 4.5. Pharmacokinetic Analysis and Statistics

For the data analysis, the microdialysate concentrations were attributed to the midpoint of the sampling interval. The pharmacokinetic parameters and statistical analysis were performed using STATA (v. 17.0 StataCorp, College Station, TX, USA) for continuous intravenous microdialysis samples and standard plasma samples separately. Standard pharmacokinetic parameters for both vancomycin and meropenem: AUC_0–8h_, C_max_, T_max_, and T_1/2_ were calculated separately for both sampling methods in all eight pigs using non-compartmental analysis. AUC was calculated using the linear-up/log-down method. C_max_ was calculated as the maximum of all the concentrations and T_max_ as the time to reach C_max_. For comparison between the two sampling methods, the difference of pharmakokinetic parameters between methods is calculated including 95% confidence intervals. The Kenward–Roger approximation method was used due to small sample size, as a correction for degrees of freedom. Model assumptions were tested by visual diagnosis of residuals, fitted values, and estimates of random effects [30].

## 5. Conclusions

Both vancomycin and meropenem concentrations obtained by intravenous microdialysis were generally lower than from standard plasma sampling, particularly in the distribution phase. The differences in key pharmacokinetic parameters between the two sampling techniques underline the importance of further investigations to find the most suitable and reliable method for continuous intravenous antibiotic concentration sampling.

## Figures and Tables

**Figure 1 antibiotics-12-00791-f001:**
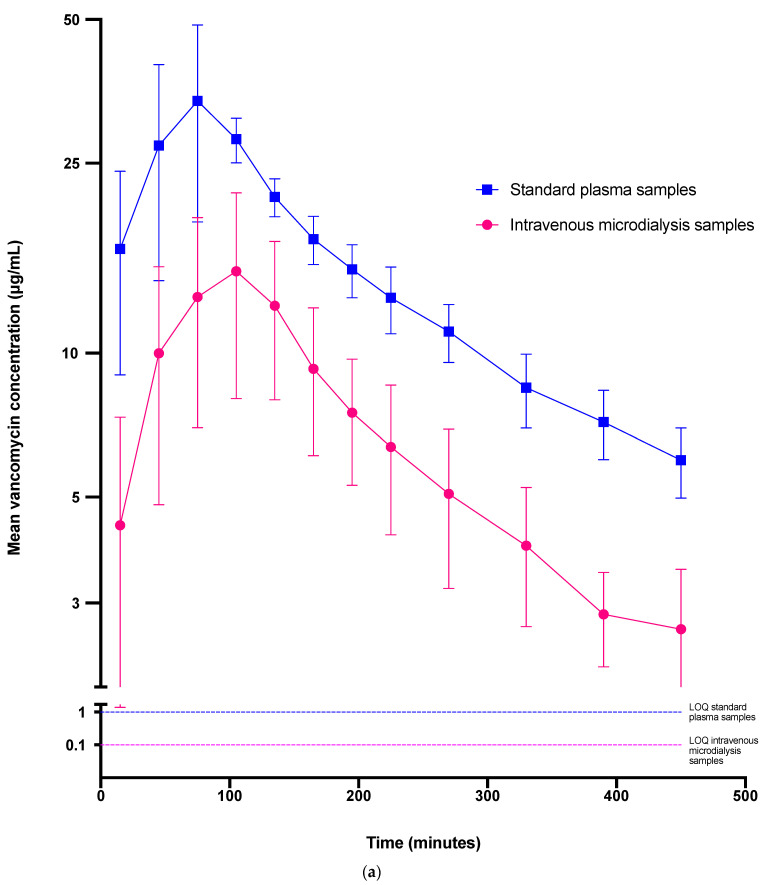
(**a**) Mean concentration–time profiles of vancomycin obtained by standard plasma samples and intravenous microdialysis samples. Standard deviation (SD) is visualized with bars. The left *Y*-axis is log-scaled and shows a two-segmented axis in order to contain all values. Lower limits of quantification (LOQ) for the analytical methods are inserted for illustration. (**b**) Individual concentration–time profiles for standard plasma sampling and intravenous microdialysis sampling in pig numbers 1–8 for vancomycin. The *Y*-axis is log-scaled.

**Figure 2 antibiotics-12-00791-f002:**
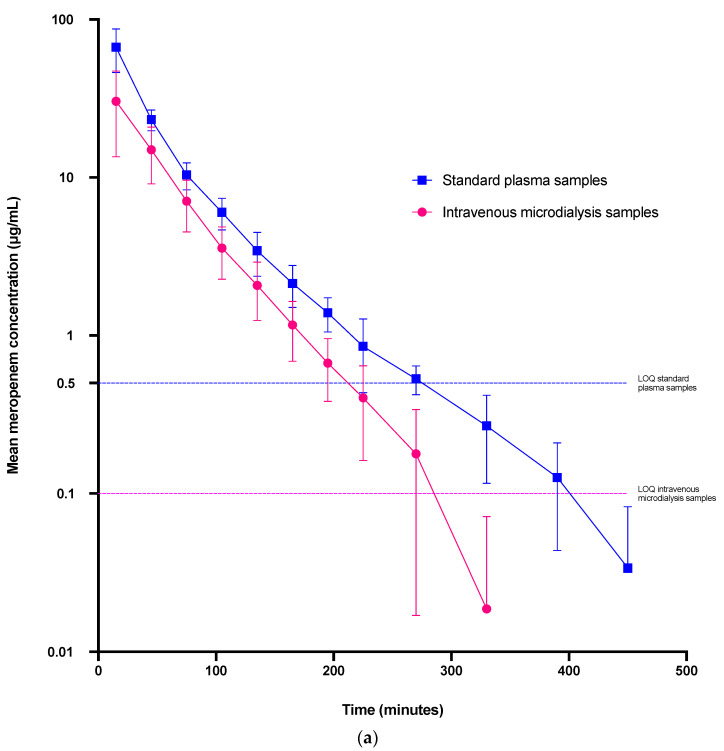
(**a**) Mean concentration–time profiles of meropenem obtained by standard plasma samples and intravenous microdialysis samples. Standard deviation (SD) is visualized with bars. The left *Y*-axis is log-scaled and shows a two-segmented axis in order to contain all values. Lower limits of quantification (LOQ) for the analytical methods are inserted for illustration. Values below this threshold should be cautiously interpreted. (**b**) Individual concentration–time profiles for standard plasma sampling and intravenous microdialysis sampling in pig numbers 1–8 for meropenem. The *Y*-axis is log-scaled.

**Figure 3 antibiotics-12-00791-f003:**
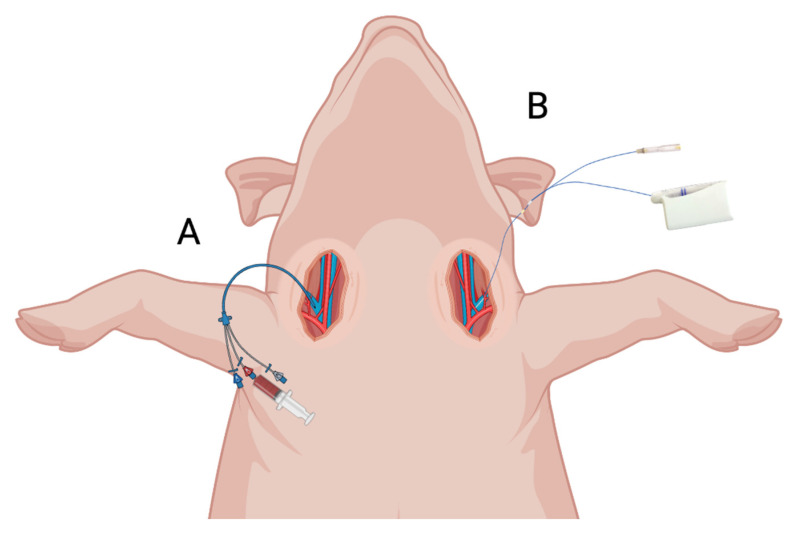
Illustrative drawing of the central venous catheter (A) and placement of the intravenous microdialysis catheter (B). Both drugs were administered through separate peripheral venous catheters placed in both ears of the pig (not illustrated in this figure). Figure 3 was made using Biorender, and permission for publication is obtained [32].

**Table 1 antibiotics-12-00791-t001:** Pharmacokinetic comparison between standard plasma samples and intravenous microdialysis samples for vancomycin.

Vancomycin	Standard Plasma Samples	Intravenous Microdialysis Samples	Difference between Methods
AUC_0–8h_, min µg/mL	7014 (5779; 8249)	2874 (1639; 4109)	−4140 (−5202; −3077)
C_max_, µg/mL	36 (24; 49)	14 (2; 26)	−22 (−35; −97)
T_1/2_, min	197 (124; 270)	242 (162; 322)	45 (−55; 146)
T_max_, min	85 (72; 98)	110 (97; 123)	25 (4.3; 46)

Values are given as means (95% confidence interval). AUC_0–8h_, area under the curve from 0 to the last measured value. C_max_, peak drug concentration. T_1/2_, half-life. T_max_, time to reach C_max_.

**Table 2 antibiotics-12-00791-t002:** Pharmacokinetic comparison between standard plasma samples and intravenous microdialysis samples for meropenem.

Meropenem	Standard Plasma Samples	Intravenous Microdialysis Samples	Difference between Methods
AUC_0–8h_, min µg/mL	2980 (2437; 3523)	1501 (958; 2044)	−1479 (−2209; 749)
C_max_, µg/mL	69 (50; 88)	29 (10; 47)	−40 (−66; −15)
T_1/2_, min	44 (39; 49)	36 (32; 41)	−7.6 (−14; 1.6)
T_max_, min	15 (7; 23)	20 (12; 28)	5 (−8; 18)

Values are given as means (95% confidence interval). AUC_0–8h_, area under the curve from 0 to the last measured value. C_max_, peak drug concentration. T_1/2_, half-life. T_max_, time to reach C_max_.

## Data Availability

The data are available on reasonable request from the authors.

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
