# Peer review of "Comparison of Intravenous Microdialysis and Standard Plasma Sampling for Monitoring of Vancomycin and Meropenem Plasma Concentrations—An Experimental Porcine Study"

_antibiotics, 2023, doi:10.3390/antibiotics12040791_

Round 1
Reviewer 1 Report
The topic of the paper is interesting and relevant as micro dialysis is a well-established method that can be a good alternative to standard plasma sampling to overcome obstacles such as PK sampling in the pediatric population. The manuscript is well written, literature has been comprehensively reviewed, clearly presented.
However, if I understand the methods correctly, there was a major mistake in the study setup:
Microdialysis only captures the free, unbound concentrations.
It seems that plasma protein binding (PPB) of vancomycin and meropenem was not determined for the standard plasma samples in this experimental setup using techniques like ultrafiltration or equilibrium dialysis.
Thereby concentration versus time profiles of mean micro-dialysate unbound concentrations of vancomycin and meropenem were compared with standard plasma samples using total concentrations.
Minor comment: Moreover, PPB is specie dependent, so, in order to extrapolate such data it is very important to first determine the PPB of pigs and compare it to humans.
Considering the issues stated above, the manuscript is not suitable of publication in the current state.
Reviewer 2 Report
Manuscript has been well written, systematic, and very concise manner, with excellent scientific evidence.
Few shortcomings have been observed, which needs author attention:
· Author should explain the standard plasma samples technique to understand the process, referred in the manuscript.
· Why Vancomycin and meropenem are used as a test drug? As Vancomycin has issue with the binding with centrifuge tube during processing, which may sometimes lead to artifact results.
· Pharmacokinetics parameters should be calculated using WinNonLin Software, to have confidence in the result. (Line no. 251)
· How much volume of blood has been withdrawn during each time point should be mentioned for both standard and Catheterized method. Is there any blood volume withdrawal different in both methods?
· Why two different methods are used for “free unbound concentrations of vancomycin in plasma were quantified using a clinical standard homogeneous enzyme immunoassay method Chemistry XPT, Advia Chemistry, Germany)” (line 236) AND “For vancomycin and meropenem, quantification of the free 244 unbound concentrations in the microdialysates was utilitzed using ultra-high performance 245 liquid chromatography and tandem mass spectrometry” (Line 244) ? Please explain section 4.4 in detail.
Is there any internal standard used for Vancomycin and Meropenem Analysis?
Reviewer 3 Report
Thank you for the request to review.
My comments concern two aspects of the manuscript:
11) The results suggest a large difference in the observations between the two sampling techniques. At face value this seems to represent bias in the microdialysis sampling method. The authors have expressed several possible reasons (138-169) for this apparent effect, including specifically some aspects of technical performance of the microdialysis method.
I think there are some other important considerations. The value of this work to the reader rests strongly on the reason for this apparent bias.
I struggle to assess this with the current graphical presentation, but it seems that concentration of vancomyin or meropenem in the microdialysis samples are essentially a constant proportion of the venous concentration. This would be plausibly explained by plasma protein binding considerations if the microdialysis concentrations represent approximately the free drug concentration, and the plasma samples the total drug concentration, presuming that the rate of drug unbinding is rate-limiting. This seems plausible presuming protein binding fraction for vancomycin of 50%, but less so for meropenem. In their analytic methodology the authors have described too little detail to assess if the plasma concentrations should be considered to represent free or total concentration. Though the method says ‘… free unbound concentration of vancomycin in plasma was quantified…’ and so on also for meropenem, they haven’t described what it is about those methods that is free-drug specific.
Based on the diagram (figure 3) it appears that there is only one venous catheter. This suggests that administration of drug and venous sampling occurred through the same device. In that case, I don’t think the authors can exclude that plasma concentrations in the study are upwards-biased by contamination from drug material in the catheter.
The reported relative recovery (62-64) of about 50% for vancomycin and about 65% for meropenem also aligns visually with the apparent constant-proportion bias observed in the graphics. I presume that there is no possibility that this apparent bias is an artefact of the data analysis strategy? The reader isn’t shown any of the data, so it is difficult to exclude the suspicion that errors in the data processing have led to the apparent bias.
22) There are a number of issues with statistical aspects of the presentation that make this needlessly difficult to read and obscure some important information.
a. Figure 1 and Figure 2 should use a logarithmic y-axis. This would make the shape of the terminal phase much easier to read and aid the direct comparison of the venous and microdialysis data. Particularly in the meropenem data so much information is compressed along the x-axis that most of the graphic is wasted.
b. The description of data using confidence intervals as in both figures 1 and 2 is inappropriate. If the purpose of these graphics is to display data, then they should do so. The confidence interval describes uncertainty, not variation, so should not be used, as in this case, to convey to the audience something about the variability in concentration. Confidence intervals from the sample mean at each time point, which I presume these are, are also simply wrong as uncertainty statements for the expected value of the concentration at any point in time, because they ignore the sequential nature of the longitudinal observations. Real uncertainty statements about concentration in PK studies are the domain of formal statistical models (i.e. population pharmacokinetics) and should be avoided in any other context.
c. Instead of the ‘mean concentration’ plot which shows no data whatsoever, the audience would be much better served by a multi-axis plot, in this case with 8 axes, each representing one subject. That way the audience is able to see the actual data, and can visually compare the actual observed concentrations between method, within-subject and within-time.
d. The phrase ‘significant difference’ and similar terms are used throughout. All usages are misleading. The authors should drop all references to ‘significance’ and focus on the substantive estimates. ‘Significance’ concepts readily generate mistakes in reasoning. For example, the authors have made such a mistake by saying ‘no statistical difference was found’ despite the actual values obviously being different (Line 84). This corresponds to misinterpretation number 6 in Greenland et al. (10.1007/s10654-016-0149-3). In the conclusions the authors report ‘The significant differences in key pharmacokinetic parameters ‘ which corresponds to misinterpretation number 7 from Greenland et al. The authors and the audience would be better served by describing the effects they observed and how uncertain they were.
e. The authors are free to present p-values if they wish, but the audience would be more informed by confidence intervals for the differences in PK parameters between conditions (rather than confidence intervals for the sample means of PK parameters, which are not correct or useful) to supplement the information in tables 1 and 2. This should be readily obtained from the linear mixed models reported by the authors already.
Round 2
Reviewer 1 Report
thank you for cosnidering my comments, I am fine with the present manuscript
Author Response
Thank you very much.
Reviewer 3 Report
Thankyou for your response.
The updates to presentation make it overall much clearer.
I have some remaining objections:
A) there are many ways to present 'data', 'inference', 'results', etc. but many of them are wrong or misleading. Regardless of tradition or its popularity, the usage of confidence intervals as summary statistics is inappropriate, because these represent inference, not data. Specifically in the case of these 'mean and error bar' graphics, because the observations from within a subject are not independent of one another, these intervals are not a sensible statement about the uncertainty of the expected concentration at any point; because they involve an assumption that is known to be violated, they cannot be expected to present reliable information. More appropriate statistics would be either the standard deviation or range, i.e. summaries of data. These are still meaningfully wrong, compared to the subject-axis graphics, but at least they present something that is closer to the data and not the result of inappropriate inference.
B) the authors state (369-373) that the vancomyin assay is a measure of free drug concentration, but don't provide any citation or data to support this claim. The only citation offered [20] is from the same authors and this doesn't include any citation or data either. Given the importance of this claim to the overall interpretation of the results I think this warrants more support.
I located some recent publications describing developments for TDM for vancomycin and, though I struggled to find any particularly clear statements, they suggest (for example, https://doi.org/10.1002/bmc.5559) that common TDM methods for vancomycin are total drug measures. Comparable studies utilized ultrafiltration (for example, https://doi.org/10.1093/jac/dkw495). I could not locate any primary evidence or even description that would support the position that the HEIA method in fact represents free drug concentration.
A few minor points:
1) the subject-axis graphics should use the same axes limits and spacing for each subject, to make it easier for the reader to compare them. Use of a common legend for the axes would also minimize the white space area around the axes and make them easier to read.
2) the confidence interval expressions in the final table have some issues. Specifically, some of them appear inconsistent with the point estimates (I suspect this is a formatting issue), and the use of '-' to separate them is easily confused with the sign '-' for negative values, making the intervals a bit difficult to read.
3) it's not clear that there are any observations, as presented in the graphics, that were below the analytic limits of quantification. The range of presented observed concentrations doesn't appear to map very directly to statements about analytic LOQ in the methods (perhaps because those are not expressed in the same scale as the plasma concentration). This would be clearer if the authors specified the number of BLOQ observations, or even better, that they specifically showed any BLOQ observations in the graphics.
